# Gut Microbiome Dysbiosis in Patients with Pemphigus and Correlation with Pathogenic Autoantibodies

**DOI:** 10.3390/biom14070880

**Published:** 2024-07-22

**Authors:** Si-Zhe Li, Qing-Yang Wu, Yue Fan, Feng Guo, Xiao-Min Hu, Ya-Gang Zuo

**Affiliations:** 1Department of Dermatology, State Key Laboratory of Complex Severe and Rare Diseases, Peking Union Medical College Hospital, Chinese Academy of Medical Sciences & Peking Union Medical College, Beijing 100730, China; lisizhe@pumch.cn (S.-Z.L.); gf720heartsoul@sina.com (F.G.); 2Department of Cardiology, State Key Laboratory of Complex Severe and Rare Diseases, Peking Union Medical College Hospital, Chinese Academy of Medical Sciences & Peking Union Medical College, Beijing 100730, China; qy-wu17@student.pumc.edu.cn (Q.-Y.W.); fanyue15@student.pumc.edu.cn (Y.F.); 3Department of Medical Research Center, State Key Laboratory of Complex Severe and Rare Diseases, Peking Union Medical College Hospital, Chinese Academy of Medical Sciences & Peking Union Medical College, Beijing 100730, China

**Keywords:** 16S rRNA, gut microbiota, pemphigus, autoantibodies

## Abstract

Background: Pemphigus is a group of potentially life-threatening autoimmune bullous diseases induced by pathogenic autoantibodies binding to the surface of epidermal cells. The role of the gut microbiota (GM) has been described in various autoimmune diseases. However, the impact of the GM on pemphigus is less understood. This study aimed to investigate whether there was alterations in the composition and function of the GM in pemphigus patients compared to healthy controls (HCs). Methods: Fecal samples were collected from 20 patients with active pemphigus (AP), 11 patients with remission pemphigus (PR), and 47 HCs. To sequence the fecal samples, 16S rRNA was applied, and bioinformatic analyses were performed. Results: We found differences in the abundance of certain bacterial taxa among the three groups. At the family level, the abundance of *Prevotellaceae* and *Coriobacteriaceae* positively correlated with pathogenic autoantibodies. At the genus level, the abundance of *Klebsiella*, *Akkermansia*, *Bifidobacterium*, *Collinsella*, *Gemmiger*, and *Prevotella* positively correlated with pathogenic autoantibodies. Meanwhile, the abundance of *Veillonella* and *Clostridium_XlVa* negatively correlated with pathogenic autoantibodies. A BugBase analysis revealed that the sum of potentially pathogenic bacteria was elevated in the AP group in comparison to the PR group. Additionally, the proportion of Gram-negative bacteria in the PR group was statistically significantly lower in comparison to the HC group. Conclusion: The differences in GM composition among the three groups, and the correlation between certain bacterial taxa and pathogenic autoantibodies of pemphigus, support a linkage between the GM and pemphigus.

## 1. Introduction

Pemphigus refers to a subset of rare and serious autoimmune dermatoses with painful erosions or blisters on the skin and mucous membranes [1,2]. The variants of pemphigus include pemphigus vulgaris (PV), pemphigus foliaceus (PF), pemphigus vegetans, pemphigus erythematosus (PE), IgA pemphigus, pemphigus herpetiformis (PH), and paraneoplastic pemphigus. Lesion formation of pemphigus is the result of pathogenic autoantibodies that target desmoglein-3 (Dsg3) and desmoglein-1 (Dsg1), located on the surface of epidermal keratinocytes [3]. The pathogenic autoantibodies are produced due to the disruption of immunological tolerance, activation of autoreactive B cells, and imbalance of T helper (Th) 1 and Th2 cells, with the activation of Th2 and T follicular helper cells being particularly important [3,4,5].

Despite recent progress in understanding the mechanisms of pemphigus, the exact cause of this condition remains unclear. Various exogenous triggers have been identified for pemphigus, such as certain drugs [6], stress [7], radiation [8], and virus infections [9,10]. However, bacterial infections are commonly seen as complications of immunosuppressive therapy for pemphigus [11,12].

The microbiota is a collection of living microorganisms that are present in a defined environmental habitat [13,14]. The impacts on healthy hosts from the gut microbiota (GM) are double-sided, affecting metabolism, disrupting the gut barrier, and regulating immunity and inflammation [14,15]. The involvement of the GM has been elucidated in various autoimmune diseases, including type 1 diabetes (T1DM) [16], rheumatoid arthritis (RA) [17], and systemic lupus erythematosus (SLE) [18]. The effects of the GM have also been reported in autoimmune dermatoses, such as psoriasis [19], vitiligo [20], bullous pemphigoid [21], and alopecia areata [22]. The association mechanisms between dermatoses and the GM are thought to involve the inflammatory immune response, which can trigger or aggravate these conditions [23]. Based on these findings, the concept of the gut–skin axis has been developed.

As a typical autoimmune dermatosis, pemphigus can reasonably be inferred to involve the GM in its pathogenesis though the gut–skin axis. However, previous studies on PV patients reveal contradictory findings regarding the composition of the GM, with one showing alterations and the other indicating a normal composition [24,25]. Two recent reports elucidated that the disturbed compositions of the GM were involved in the pathogenesis of pemphigus [26,27]. Our research contributes substantiating evidence regarding the correlation between the GM and pemphigus to investigating the underlying mechanisms and potential approaches for the disease.

## 2. Materials and Methods

### 2.1. Patient Recruitment and Grouping

All the pemphigus patients were recruited from the department of Dermatology in Peking Union Medical College Hospital (PUMCH). The PUMCH’s Ethical Review Committee provided our study with ethical approval. The identifier number is JS-2562. All the patients recruited to this study signed the informed consent forms. Pemphigus was diagnosed according to the following guidelines [2]: (i) clinical history collection and physical examination conduction to recognize the characteristic symptoms and signs of pemphigus; (ii) histopathological examination to observe typical presentation; (iii) direct immunofluorescence (DIF) examination of a perilesional skin or mucosal biopsy to detect typical immunological patterns; (iv) autoantibodies against the epithelial cell surface detection using serological methods, such as indirect immunofluorescence (IIF) and/or an enzyme-linked immunosorbent assay (ELISA), specifically targeting Dsg1/3 IgG.

The exclusion criteria for all the recruited subjects were as follows: (i) no history of severe autoimmune diseases, including SLE, dermatomyositis, systemic sclerosis, multiple sclerosis, and RA; (ii) no history of infectious diseases that could potentially impact the composition of the GM, such as acute gastrointestinal infection, bacteremia, active tuberculosis, or a history of HIV infection within two months preceding the sampling, or (iii) no history of malignancy and intestinal diseases.

The active pemphigus (AP) group comprised patients in an active stage. We did not limit our inclusion criteria during the collection stage to only previously untreated, relapsed, or undertreated patients because of their rarity. Instead, we focused on patients with an active condition, regardless of whether there was recurrence or not. The pemphigus remission (PR) group consisted of patients in a remission stage, characterized by the absence of new or established lesions, while they were receiving minimal therapy according to the Recommendations of the International Panel of Experts [28]. The majority of HCs were the spouses of the patients, in order to minimize the impact of dietary structure, lifestyles, and living environments on the PV patients.

### 2.2. Clinical Data and Sample Collection

After obtaining informed consent, we recorded the participants’ demographic data (gender, age) and clinical information (past medical history, physical examination findings, and laboratory results). The recorded laboratory results included dermatopathology, DIF, IIF, and anti-Dsg1/3 IgG titers. Treatment modalities comprised oral and/or intravenous administration of corticosteroids and other immunosuppressive agents, and the treatment duration was also recorded simultaneously.

The participants provided fecal samples either on the same day or the following day of their visit. They received 2 ml sterile phials (containing 8% dimethyl sulfoxide buffer, 2% guanidine isothiocyanate, 0.4% ethylenediaminetetraacetic acid, 0.5% Tris(hydroxymethyl)aminomethane hydrochloride, and 0.1% bromothymol blue), accompanied by particular illustrated instructions for sample collection. The freshly collected fecal samples were then either frozen at −80 °C or transported to the laboratory for sequencing.

### 2.3. DNA Extraction and 16S rRNA Gene Sequencing

Utilizing the PowerSoil DNA Isolation Kit (made by MoBio Laboratories, Carlsbad, CA, USA), bacterial DNA was extracted from the fecal samples following the manufacturer’s instructions. We then assessed the integrity and size of the DNA using gel electrophoresis on 1% agarose gels. Additionally, we subjected the amplicons (with the upstream primer 5′-ACTCCTACGGGAGGCAGCAG-3′ and the downstream primer 5′-GGACTACHVGGGTWTCTAAT-3′) to a sequence analysis and performed a raw data quality check [29]. The sequencing library was constructed for the 16S rRNA gene’s V3 and V4 regions [30]. The purified products were combined in an equimolar ratio, and subsequent purification was performed using the AxyPrep Gel Extraction Kit (made by Axygen Biosciences, Swedesboro, NJ, USA). We generated sequencing libraries using the NEBNext Ultra DNA Library Prep Kit for Illumina (made by New England Biolabs, Ipswich, MA, USA), following the manufacturer’s recommendations, which included the addition of index codes. To assess the library quality, we utilized the Qubit 2.0 Fluorometer (made by Thermo Fisher Scientific, Waltham, MA, USA) and the Agilent Bioanalyzer 2100 system (made by Agilent Technologies, Santa Clara, CA, USA). Finally, library sequencing was conducted using the IlluminaMiSeq PE300 platform (made by Illumina, San Diego, CA, USA).

The usage of EasyAmplicon helped us to conduct the downstream amplicon analyses [31]. Then, we also employed the VSEARCH software package (Version 2.15.0) to merge paired reads, label samples, cut primers, perform quality filtering, and dereplicate the sequences [32]. Subsequently, these unique sequences underwent denoising into amplicon sequence variants (ASVs) using the unoise3 algorithm implemented in USEARCH (Version 10.0). An ASV table was generated using the vsearch—usearch_global function. Taxonomic classification of ASVs relied on the sintax algorithm within USEARCH on the basis of the Ribosomal Database Project (RDP) training set (Version 16) [33]. Normalization of ASVs was achieved using the otutab_stats algorithm in USEARCH. Finally, the resulting ASV table facilitated subsequent statistical analyses, including assessments of sufficient taxa and alpha and beta diversity differences.

### 2.4. Bioinformatics Prediction and Statistical Analysis

R software (Version 4.0.2) was employed for analyzing the microbial diversity. Alpha diversity analyses were conducted using the vegan package (Version 2.5-6). Beta diversity was calculated using a principal coordinate analysis (PCoA) in accordance with the Bray–Curtis distance of microbial communities, which extracted the maximum classification subplane for display. To illustrate ASV overlapping between groups, Venn diagrams were generated using the VennDiagram package. Additionally, rarefaction curves were created using vsearch-alpha_div_rare. The ggplot2 package was used for the visualization of the diversity analyses and rarefaction curves. At the respective family and order levels, the microbial community compositions were shown as stacked bar plots among three groups. To explore the relationship between different groups at the taxonomic level of genus or phylum, chord diagrams were constructed using the tax_circlize package. Furthermore, Volcano plots and Manhattan plots revealed differential ASVs between groups using various R packages. For assessing differences in ASV abundance among groups, we utilized the edgeR package, and the Benjamini–Hochberg method was employed to control the false discovery rate [34]. Heatmap results were generated using SPSS (Version 25.0) and the pheatmap package. We chose the top 20 differential bacteria in the family level and all in the genera level. To predict the phenotypes and functional pathways of the bacterial community, we employed BugBase and Phylogenetic Investigation of Communities by Reconstruction of Unobserved States 2 (PICRUSt2, https://github.com/picrust/picrust2/ (accessed on 30 June 2024) [35], respectively. Finally, STAMP (Version 2.1.3) facilitated multi-group comparisons and generated post-inspection charts.

## 3. Results

### 3.1. Characteristics of the Study Population

Between November 2016 and May 2022, our investigation included a cohort of 31 individuals diagnosed with pemphigus, with 20 patients in the AP group and 11 patients in the PR group. The AP group encompassed 15 patients diagnosed with PV, along with 2 cases each of PF, PE, and 1 case of PH. The PR group comprised seven PV patients, three PE patients, and one PH patient. Additionally, we enrolled 47 HCs. Further details about the basic characteristics of these three groups are presented in Table 1.

### 3.2. The Composition of Microbiota Varied among the AP, PR, and HC Groups

To investigate the diversities of the GM in the pemphigus patients, the microbial community in the stool samples of both the HCs and pemphigus patients were analyzed using 16S rRNA sequencing. The ACE diversity and Chao 1 indices were applied to assess the microbial community richness of alpha diversity, while the Shannon and Simpson indices were used to measure the microbial community evenness. Although no significant differences were observed among the four indices of alpha diversity, the microbial community richness progressively decreased from the HC group to the PR group and then to the AP group, as indicated by the ACE (*p* = 0.612) and Chao1 (*p* = 0.584) indices (Figure 1A,B). The Shannon (*p* = 0.603) and Simpson (*p* = 0.615) indices of the HC group exhibited marginally decreased values compared to the other groups, but these differences did not attain statistical significance (Figure 1C,D). Beta diversity was assessed using PCoA, but no noticeable distinction was observed in the microbial communities among the three groups in the scatter plot (Figure 1E). Additionally, a Venn diagram illustrated the presence of amplicon sequence variants (ASVs), with a relative abundance of >0.1% in each group (Figure 1F). All three groups shared 80 ASVs, while the AP group vs. PR group, AP group vs. HCs group, and PR group vs. HCs group had 11, 23, and 17 shared ASVs, respectively. Additionally, the AP group, PR group, and HC group had 46, 50, and 41 ASVs, respectively, that were not present in the other groups. The adequacy of the sequencing data was confirmed by analyzing the rarefaction curves (Appendix A).

### 3.3. Compared to HCs, Pemphigus Patients Exhibited an Altered Abundance of GM Taxa

Regarding taxonomy, the dominant microbiota composition was similar across the different groups at every level, but the relative abundance varied. Firmicutes and Bacteroidetes were identified as the dominant phyla at the phylum level (Figure 2A). The Firmicutes/Bacteroidetes (F/B) ratio, often used as a marker for inflammatory disease, presented a decreasing trend in the PR group and an increasing trend in the HC group, but these variances did not achieve statistical significance (AP group: 1.15, PR group: 0.98, HC group: 1.39, *p* = 0.456). At the order level, the PR group had a significantly lower relative abundance of *Burkholderiales* compared to the HC group (Figure 2B). At the family level, the PR group presented the highest relative abundance of *Lachnospiraceae*, whereas the AP group exhibited the lowest relative abundance of *Veillonellaceae*. The relative abundance of *Prevotellaceae* was observed to increase progressively in the AP, HC, and PR groups at the family level, though these differences level were not statistically significant (Figure 2C). At the genus level, the relative abundance of *Blautia* was significantly higher in the PR group compared to the AP groups. The relative abundance of *Prevotella* showed a progressive increase across the AP, HC, and PR groups, although this trend was not statistically significant (Figure 2D).

### 3.4. Patients in the Active Stage Showed Significant Alterations in GM Abundance Compared to Those in the Remission Stage at the Species Level

We used edgeR (*p* < 0.05 and FDR < 0.2) to compare the differences in the GM between the groups. Between the AP and HC groups, a total of 37 ASVs presented significantly different abundances, with 11 ASVs enriched in the AP group and 26 in the HC group (Figure 3A). Compared to the PR group, the AP group showed 35 ASVs with significantly different abundances (Figure 3B), with 20 ASVs enriched and 15 ASVs depleted in the AP group. Between the PR and HC groups, only 18 ASVs showed different abundances, with 17 ASVs depleted and one ASV enriched in the PR group (Figure 3C).

To demonstrate the contributions of differentially abundant ASVs at the phylum level between the AP and PR groups, a Manhattan plot was used (Figure 3D). The enriched ASVs in the AP group mainly belonged to Bacteroidetes and Proteobacteria, while the depleted ASVs were primary in Firmicutes when compared to the PR group. The ASVs with different relative abundances among the three groups are shown in Appendix A.

### 3.5. The GM Was Found to Be Correlated with Clinical Indicators in Patients with Pemphigus

The immune serological tests, including detection by IIF or titers of anti-Dsg1/3 IgG, were correlated with the extent and activity of the disease [2]. We sought to correlate the serological test results with the top 20 differential bacteria at the family level, as well as the between-group differential genera and ASVs. At the family level (Figure 4A), the abundances of *Prevotellaceae* (*p* = 0.044) and *Coriobacteriaceae* (*p* = 0.039) were positively correlated with anti-Dsg1 IgG titers. At the genus level (Figure 4B), the abundance of *Klebsiella* was positively correlated with anti-Dsg3 IgG titers (*p* = 0.049), while the abundance of *Veillonella* was negatively correlated (*p* = 0.045). The abundances of five genera were positively correlated with anti-Dsg1 IgG titers, including *Akkermansia* (*p* = 0.007), *Bifidobacterium* (*p* = 0.032), *Collinsella* (*p* = 0.036), *Gemmiger* (*p* = 0.027), and *Prevotella* (*p* = 0.021). The abundance of *Clostridium_XlVa* was negatively correlated with anti-Dsg1 IgG titers (*p* = 0.002). There were no significant correlations found between the abundances of any genus and IIF titers.

At the species levels (Appendix A), the abundances of four ASVs, including ASV_12 (*Faecalibacterium prausnitzii*, *p* = 0.005), ASV_42 (*Bifidobacterium, Unassigned*, *p* = 0.04), ASV_68 (*Faecalibacterium prausnitzii*, *p* = 0.005), and ASV_71 (*Alistipes onderdonkii*, *p* = 0.005), exhibited a positive correlation with anti-Dsg3 IgG. The abundance of ASV_12 (*Faecalibacterium prausnitzii*, *p* = 0.04) was positively correlated with anti-Dsg1 IgG, and the abundance of ASV_68 (*Faecalibacterium prausnitzii*, *p* = 0.005) was positively correlated with IIF titers.

### 3.6. Predictive Functional Pathways of the Microbial Community Were Analyzed in Patients with Pemphigus and HCs

Utilizing the BugBase database, we predicted the phenotypes of the GM in each group. Remarkably, in comparison to the PR group, the AP group exhibited a significantly elevated proportion of potentially pathogenic bacteria (*p* = 0.044, Figure 5A). The proportion of Gram-negative bacteria in the PR group was significantly reduced, in contrast to the HC group (*p* = 0.029, Figure 5C).

Using PICRUSt2 with reference to the Kyoto Encyclopedia of Genes and Genomes database, we observed differences in the microbial functional pathways among the three groups. The AP group had four pathways that significantly differed from the HC group, including pyruvate metabolism, arginine and proline metabolism, and phosphonate and phosphinate metabolism, as well as eight pathways that were more frequent in the HC group (Figure 6A). When comparing the PR group to the AP group, the differences in 17 pathways were higher and the differences in 11 pathways were lower in the AP group (Figure 6B). Furthermore, we found higher levels of 29 pathways in the HC group and 5 pathways in the PR group (Figure 6C).

## 4. Discussion

GM dysbiosis has been reported in many autoimmune diseases, such as SLE [18], and T1DM [16], inflammatory bowel disease [36], and some autoimmune dermatoses such as psoriasis [19] and vitiligo [20]. However, even in the same disease, microbial diversity can vary greatly. In pemphigus, Scaglione et al. [25] did not report significant differences in patients with PV. Our study found that the alpha diversities were similar among the three groups, which confirms prior studies [24,26,27]. Previous studies indicated alterations in the compositions of the GM in pemphigus patients in the beta diversity analysis [26,27]. However, Huang et al. [24] did not find a significant difference in the beta diversity of the GM between pemphigus patients and HCs. Our investigation did not present a significant separation among the three groups. The discrepancies between different studies may be due to variations in the participants and limited sample sizes.

The dominance of Firmicutes, Bacteroidetes as the primary phyla was consistent across all the study groups, which was consistent with observations of the guts of healthy humans and some patients [14,25]. A lower F/B ratio may be associated with the inflammatory process of the disease [37]. In our study, we also observed a slightly decreased F/B ratio in the AP patients and a slight elevation in the patients with relieved disease. Gou et al. [27] reported an increase in the relative abundances of *Proteobacteria* and *Verrucomincrobia*, with a depletion of *Firmicutes* in pemphigus patients. While we did not observe the same differences, the Manhattan plot in our study revealed that the most depleted ASVs belonged to *Firmicutes* in the active pemphigus patients, and the enriched ASVs predominantly belonged to *Bacteroidetes* and *Proteobacteria*. These findings indicate that dysbiosis of GM may serve as a pathogenic factor in pemphigus and suggest the potential therapeutic mechanisms of GM regulation as a treatment approach for pemphigus.

Furthermore, we analyzed the specific taxa in the three groups and their association with the titers of pathogenic autoantibodies. Others have observed decreases in the relative abundance of *Veillonellaceae* in the intestines of patients with juvenile idiopathic arthritis [38], Graves’ Disease [39], and eczema [40]. In contrast, a relatively higher abundance of *Veillonella* has been reported in patients with autoimmune hepatitis [41], T1DM [42], and psoriasis [43]. Our study found a significant reduction of *Veillonellaceae* and *Veillonella dispar* (ASV_97) in the AP group, with the relative abundance of *Veillonella* displaying a negative correlation with the anti-Dsg3 IgG titers. *Veillonella* species have been found to promote the Th1 immune response and modulate the Th2 immune response [44], which may account for the results. The *Prevotellaceae* family and *Prevotella* genus have been reported to be enriched in patients with Graves’ disease [39] and RA [45] but depleted in autoimmune hepatitis [46]. In our investigation, we observed a depletion of *Prevotellaceae* and *Prevotella*, which were positively correlated with anti-Dsg1 IgG titers, in the AP patients. This may explain the paradoxical finding that different species of *Prevotellaceae* and *Prevotella* have opposing effects on T lymphocyte polarization [47,48,49]. In addition, we observed that five ASVs from *Prevotella copri* were depleted, but another ASV was enriched in the AP group. Previous research has suggested that *P. copri* may be a complex comprising four genetically distinct clades [50]. Considering the previous studies and our findings, we suggest that *Prevotellaceae* and *Prevotella* may play a protective role in pemphigus by promoting Th1 response and balancing Th1/Th2 [48,51]. Additionally, the *Clostridium_XlVa* genus, whose relative abundance was found to correlate with pathogenic autoantibodies titers, and a species (ASV_113) that was depleted in the AP group, may also be associated with protective effects.

The *Faecalibacterium prausnitzii* showed positive correlations with pathogenic autoantibodies titers, and the *Faecalibacterium* genus exhibited a slight increase in the AP group. The results imply a plausible relationship between *Faecalibacterium* and pemphigus. Previous studies indicate that *Faecalibacterium* is depleted in patients with several autoimmune diseases [52]. *Faecalibacterium* and *F. prausnitzii* are known probiotic bacteria that have anti-inflammatory properties and release butyrate [52,53]. Interestingly, previous studies have shown opposing results, where butyrate can promote inflammation and increase disease severity in RA [54]. These opposing results suggest a dual role for *Faecalibacterium* in autoimmune diseases. Similarly, *Bifidobacterium*, a well-known probiotic bacterium, was correlated with anti-Dsg3 IgG titers at the genus and species levels in our study. Different species and strains of *Bifidobacterium* can induce diverse immune responses [55]. Previous studies have implied that *Bifidobacterium* has causal effects on T1DM and celiac disease [56]. *Akkermansia*, which is also considered to be a probiotic bacterium, was recently found to enhance inflammation in multiple sclerosis [57]. Similarly, the *Akkermansia* genus showed an association with pathogenic antibodies of pemphigus in our observation. In general, bacteria that are considered probiotic may be positively associated with pemphigus and some other autoimmune diseases.

Our study also found that the *Coriobacteriaceae* family, as well as the genera *Collinsella*, *Gemmiger*, and *Klebsiella* and the species *Alistipes onderdonkii*, were positively correlated with titers of pathogenic autoantibodies. The *Coriobacteriaceae* family was found to be elevated in a mouse model of RA and was shown to affect immune reconstitution [58,59]. This positive correlation between *Klebsiella* and the titers of pathogenic autoantibodies was similarly reported in a previous study [27]. These bacteria were also reported to be enriched in patients with autoimmune diseases and dermatoses [17,60,61]. Nevertheless, a contradictory correlation was observed between a previous study and our finding. In a prior report, *Gemmiger*, *P. copri* and *Bifidobacterium* were negatively correlated with pathogenic autoantibodies level [27]. Contradictory findings regarding the relationship between the GM and pathogenic autoantibodies were also reported in previous studies. In a study by Wang et al. [26], a positive correlation was reported between *Lachnospiraceae bacterium* and anti-Dsg antibody, whereas a negative correlation was reported in a study by Guo et al. [27]. These conflicting results suggested the necessity of further experimental confirmation of the function of specific species of the GM, emphasizing the need for a more cautious approach in regulating intestinal microflora on pemphigus.

Furthermore, we conducted predictions of both phenotypes and functional pathways within the GM across the three groups. Our investigation revealed that the patients in the AP group had a higher proportion of potentially pathogenic bacteria, whereas the individuals in the HC group exhibited a higher prevalence of Gram-negative bacteria compared to the PR group. These findings suggest that alterations in the GM’s composition and metabolism result not only from pemphigus but also from pemphigus treatments, such as corticosteroids and immunosuppressive agents. Additionally, the GM of the pemphigus patients showed aberrations in pyruvate metabolism, arginine and proline metabolism, and phosphonate and phosphinate metabolism, which include carbohydrate and amino acid metabolism. These differential functional pathways have been reported to regulate inflammatory and immune responses and affect autoimmune diseases [62,63,64]. Another study on the GM of pemphigus patients implied that the GM influences the mechanisms of pemphigus through its effect on T cell differentiation [24]. It is hypothesized that the GM influences metabolic pathways to regulate immune responses and participates in the pathogenesis of pemphigus, as current findings are purely observational. Therefore, further research is required to fully elucidate the role of the GM in the pathogenesis of pemphigus.

This study has some limitations. First, some patients received systemic corticosteroids and immunosuppressive agents when fecal samples were collected. Additionally, variations in dietary habits could be a confounding factor, even though we enrolled the spouses of the patients as a part of the control group. Second, the sample size of this research is limited, which could potentially obscure weak correlations between pemphigus and the GM. As pemphigus is a rare disease, it is challenging to enroll a larger number of patients. Meanwhile, we did not conduct separate studies on different types of pemphigus. Moreover, the ages of the three groups were significantly different. Age-related alterations in the GM are substantial and may act as a confounding factor in our study [65]. These limitations could also explain the differences between previous studies and our findings [24,25].

In conclusion, the GM profile showed differences between the pemphigus patients and HCs, suggesting a unique microbial signature in pemphigus. Our study also revealed an association between certain bacterial taxa and the pathogenic autoantibodies of pemphigus.

## Figures and Tables

**Figure 1 biomolecules-14-00880-f001:**
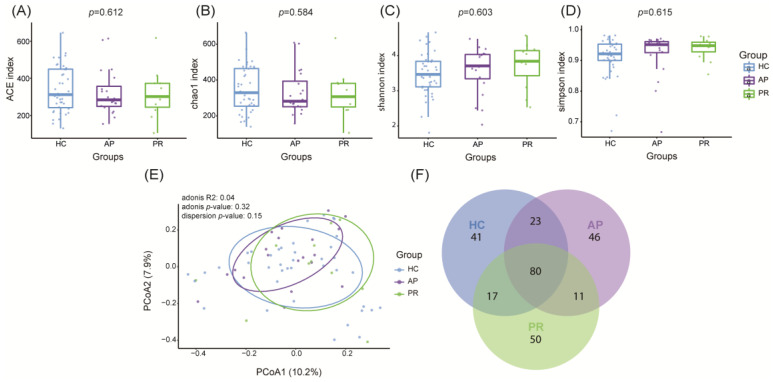
Alpha and beta diversity analysis among the AP (n = 20), PR (n = 11), and HC (n = 47) groups. (**A**–**D**) Alpha diversity based on a one-way analysis of variance presented by ACE index (**A**), Chao1 (**B**), Shannon index (**C**), and Simpson index (**D**). (**E**) Beta diversity analysis based on principal coordinate analysis (PCoA) plot; (**F**) A Venn diagram demonstrating the existence of ASVs in each group. *p* values in (**A**–**D**) indicate the differences in ANOVA correlations among the three groups. ASV, amplicon sequence variants; AP group, active pemphigus group; PR group, pemphigus remission group; HC group, healthy control group.

**Figure 2 biomolecules-14-00880-f002:**
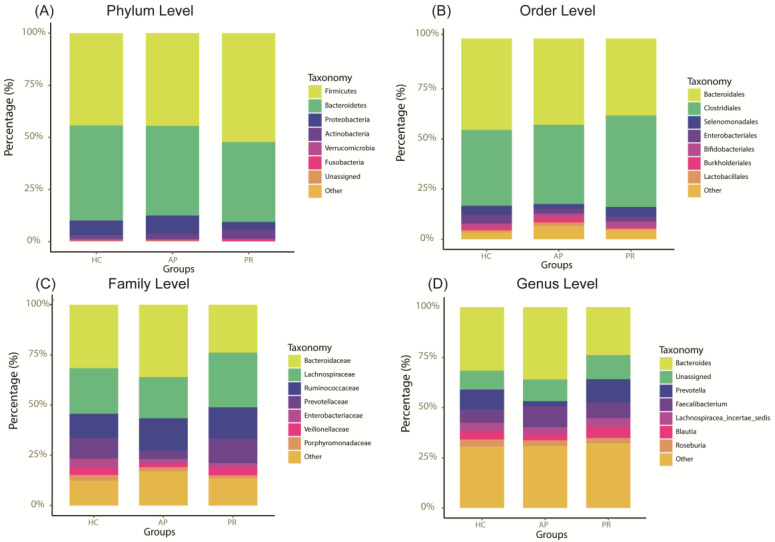
The illustration of taxonomic features of the AP, PR, and HC groups. (**A**–**D**) Relative abundances of bacteria among the three groups at the phylum (**A**), order (**B**), family (**C**), and genus (**D**) levels. AP group, active pemphigus group; PR group, pemphigus remission group; HC group, healthy control group.

**Figure 3 biomolecules-14-00880-f003:**
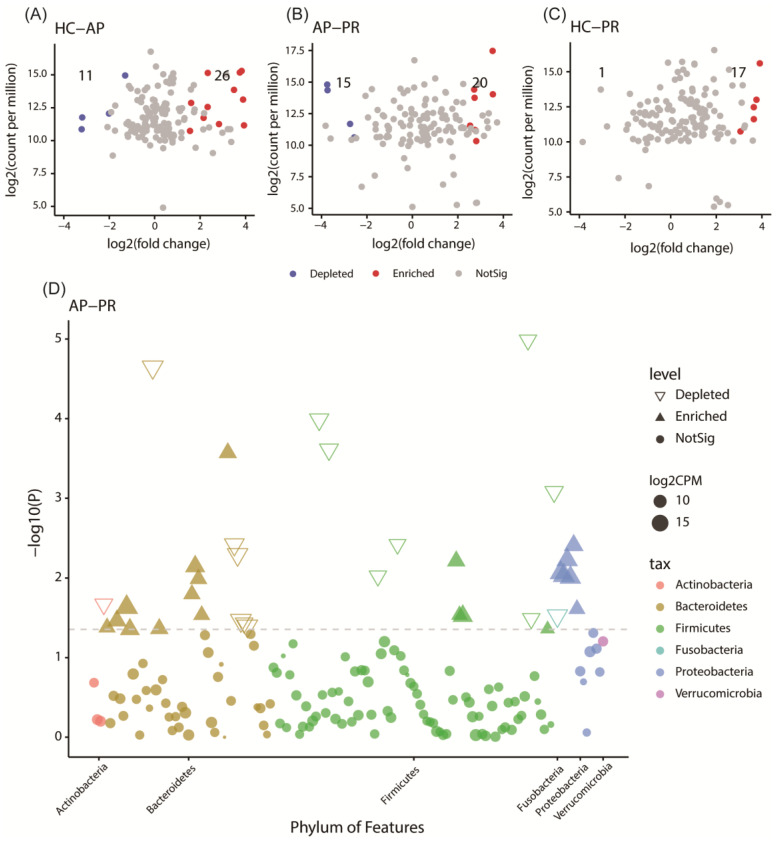
Differential ASV depiction among the AP, PR, and HC groups. (**A**–**C**) Volcano plots generated using the edgeR package. Each point represents an ASV. Significantly different ASVs are color-coded (e.g., AP vs. PR, blue = depleted in the AP group; red = enriched in the AP group; gray = not significant). (**D**) The Manhattan plot of differential taxa between the PV uncontrolled and PV remission groups. The x-axis represents microbial ASV taxonomy at the phylum level, ranked alphabetically, while the y-axis displays -log10 (*p* value). Filled triangles, hollow inverted triangles, and solid circles correspond to enriched, depleted, and insignificantly different ASVs, respectively. Each marker’s color reflects the taxonomic affiliation of the ASVs, and its size corresponds to the relative abundances using log2-transformed CPM values. ASV, amplicon sequence variants; CPM, count per million. AP group, active pemphigus group; PR group, pemphigus remission group; HC group, healthy controls group.

**Figure 4 biomolecules-14-00880-f004:**
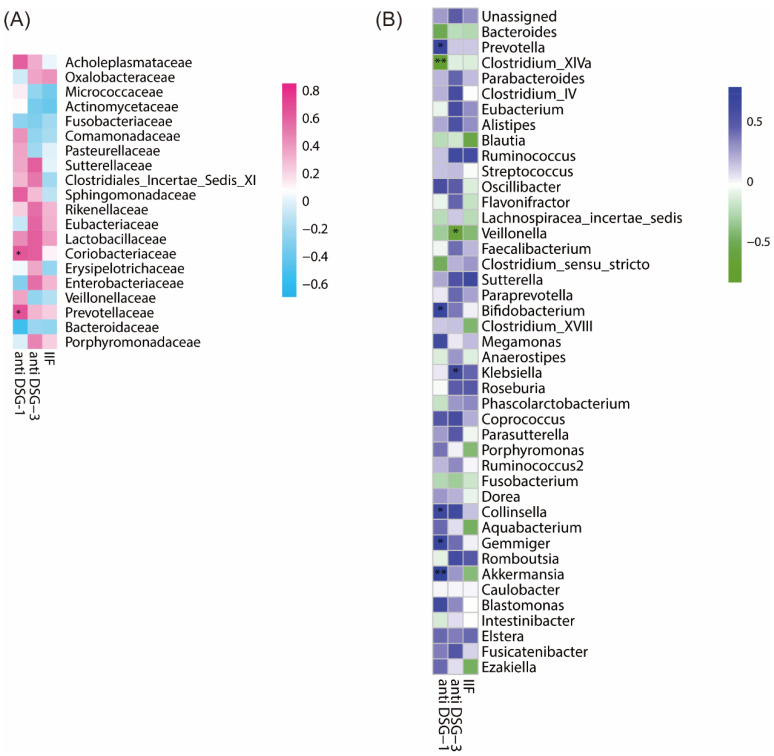
Between-group differential families and genera, along with their correlations with clinical parameters. (**A**) Spearman correlations between the top 20 differential families and clinical indicators, where *p* < 0.01 is denoted by ** and *p* < 0.05 by *. (**B**) Spearman correlations between differential genera and clinical indicators using the same significance notation.

**Figure 5 biomolecules-14-00880-f005:**
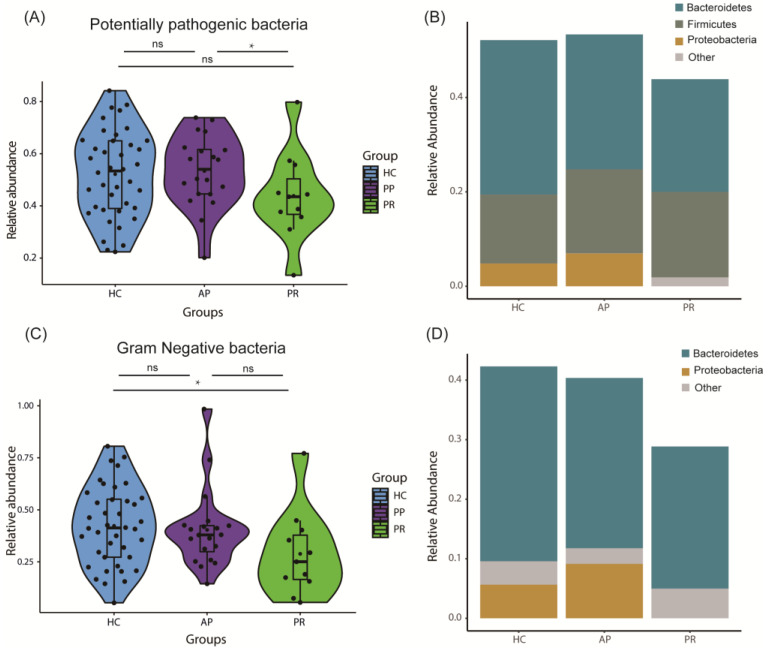
The predicted phenotypes of gut microbiota in the AP, PR, and HC groups. (**A**) Relative abundance of potentially pathogenic bacteria predicted in accordance with the BugBase database. *p* < 0.05 by *. (**B**) Distribution of potentially pathogenic bacteria at the phylum level within each group. (**C**) Relative abundance of Gram-negative bacteria predicted using the BugBase database. (**D**) Distribution of Gram-negative bacteria at the phylum level across the different groups. AP group, active pemphigus group; PR group, pemphigus remission group; HC group, healthy control group.

**Figure 6 biomolecules-14-00880-f006:**
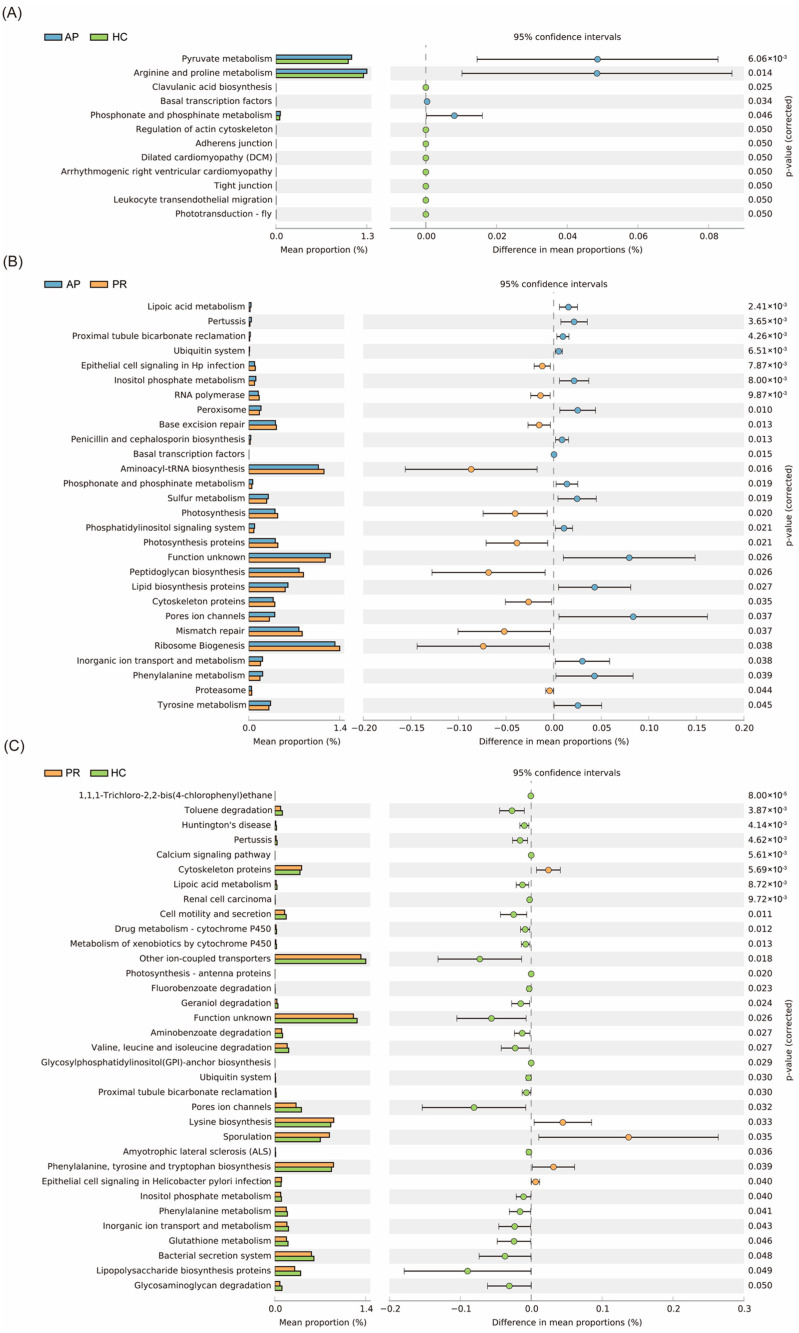
Differential functional pathways of bacteria in three groups predicted by PICRUSt2. (**A**) AP group vs. HC group; (**B**) AP group vs. PR group; (**C**) PR group vs. HC group. All pathways represented here were analyzed using a double-sided Welch’s *t*-test with a significant *p* value of <0.05. AP group, active pemphigus group; PR group, pemphigus remission group; HC group, healthy control group.

**Table 1 biomolecules-14-00880-t001:** List of study participants.

	AP Group †	PR Group ‡	HCs Group §	*p*-Value
	(n = 20)	(n = 11)	(n = 47)	
Age, years	52.80 ± 16.79	60.36 ± 12.31	62.62 ± 11.45	0.024
Female	11 (55.00%)	6 (54.55%)	29 (61.70%)	0.834
Anti-Dsg1 ¶	125.32 ± 55.15	76.91 ± 74.34	NA ††	0.064
Anti-Dsg3 ¶	82.95 ± 7996	78.27 ± 75.63	NA	0.620
Subtypes ‡‡				
PV	15 (75.00%)	7 (63.64%)	NA	
PF	2 (10.00%)	0	NA	
PE	2 (10.00%)	3 (27.27%)	NA	
PH	1 (5.00%)	1 (9.09%)	NA	
Systemic therapy				
Untreated	9 (45.00%)	0		
Systemic corticosteroids	5 (25.00%)	7 (63.64%)		
Systemic immunosuppressants	1 (5.00%)	1 (9.09%)		
Systemic corticosteroids + immunosuppressants	5 (25.00%)	3 (27.27%)		
Dosage of systemic corticosteroids (prednisolone or equivalent) §§	33 ± 22.51	8.18 ± 3.18	NA	0.007

A one-way analysis of variance and the Bonferroni test were used to compare the ages of the participants among the groups. The sex proportions were compared using the χ2 test, while the levels of anti-Dsg1 and anti-Dsg3 between the groups were compared using the Student *t* test. † AP: active pemphigus; ‡ PR: pemphigus remission; § HCs: healthy controls; ¶ Dsg: desmoglein; †† NA: not available; ‡‡ PV: pemphigus vulgaris; PF: pemphigus foliaceus; PE pemphigus erythematosus; PH: pemphigus herpetiformis. §§ The dosage of systemic corticosteroids of the AP group only including the patients treated with corticosteroids, excluding the untreated patients.

## Data Availability

The data presented in this study are available from the corresponding author upon request, as there is a slight risk of compromising the privacy of individual patients.

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
