# Peer review of "Gut Microbiome Dysbiosis in Patients with Pemphigus and Correlation with Pathogenic Autoantibodies"

_biomolecules, 2024, doi:10.3390/biom14070880_

Round 1

Reviewer 1 Report

Comments and Suggestions for Authors

Li and coauthors performed the characterization of the gut microbiota in patients with active or remission pemphigus documenting findings supporting the existence of a link among intestinal microbiota and pemphigus pathogenesis.

Major comments:

-The authors have to better explain in the introduction the importance of GM in cutaneous diseases, and specifically in pemphigus, in order to better define the aims of their study.

-Since one of the main factors influencing the GM composition is age, the authors have to add in the limits of the study the significantly different ages of the enrolled patients and controls.

-In the discussion section the authors have to discuss the mechanism by which intestinal bacteria could play a role in pemphigus, for instance referring to their metabolites and the presence of a gut-skin axis

Minor comments:

-please rewrite the sentence at lines 60-62

-please better define the DMSO buffer that you have used for sample collection

-Is possible to prepare a supplementary table including the taxa corresponding to the enriched/depleted ASVs in Figure 3 A-B-C

Comments on the Quality of English Language

The paper is in general well written and only a few modification could be made to increase the fluency of the manuscript

Reviewer 2 Report

Comments and Suggestions for Authors

 This study aims to describe gut microbiota composition in individuals with pemphigus compared to healthy controls. A positive aspect is the presence of ethics approval for the sample collection. Another good point is the use of co-housed individuals as the "healthy control" population, as it minimizes the impact of diet, lifestyle, and environment. This is a noteworthy approach, as it is not always feasible in similar studies. Methods are generally well written, and precise enough for reproducibility. It is quite unclear why there is an extensive narrative on the pathophysiology of pemphigus in introduction section, including Staph aureus toxins, when it is unrelated to the main point of this study: a 16S metabarcoding description.

I don't think the attempt to perform a correlative analysis between differentially abundant taxa and clinical parameters, as described in Table 1, is well represented in Figure 4. Substantial revisions are suggested. First, why not filter to species that are differentially abundant? This would simplify the heatmap and improve the significance of these correlations. The full dataset heatmap could still be included as a supplementary figure. What is the point of panel B? There are no significant correlations at the ASV level, which (ASV) essentially means nothing but a specific fasta sequence. A recurring comment I have for these types of studies is that 16S metabarcoding using the USEARCH pipeline is not robust below the Family level. There is a high likelihood of multi-affiliation at the genus level, so I typically avoid emphasizing ranks below the Family level. Instead of using ASV, I would recommend performing spearman correlations at the Family level (Top 20 or Top 15). This would provide more meaningful and robust results.

I understand the rationale behind Picrust/bugbase. However, I generally advise against spending too much time on these data, as they are typically poorly informative with 16S dataset without subsequent experimental validation. That said, I don't see any major flaws in this approach and will leave it to the interested community to access these data. One question, though: it is unclear from the method how the taxonomic assignments on RDP were transformed into a Picrust2-compatible one(Greengenes). This transformation would involve some manipulation for ambiguous or multi-affiliated taxa, which could influence the robustness of the findings.

The figure legends generally lack indications of the statistical tests used and the cohort size per group. In Figure 1, where are the p-values? It says "not statistically significant," but what test was exactly performed? In panel 1E for the PCoA, there is a p-value of 0.83. What exactly was performed here? Remark:  I am much surprised by this value when looking at the ellipses.

Figure 1G is rather a quality control than purely informative. You could keep as suplementary information for the interested reader.

Figure 2. It would be beneficial to reorder panels from the highest to lowest taxonomic rank. We like to read left to right, top to bottom.

Figure 3. What does it mean the numbers in the volcano plots? You need to specify somewhere (in addition to methods only) which stat test has been used to qualify those color dots as significant. Instead of "depleted" and "enriched" it would be more explicit to add "enriched in group XX" vs "enriched in group YYY". Panel D is too complex as is and we cannot clearly grasp the main information from that choice. Please clarify.

PCoA or CPCoA : am I missing something here?

Comments on the Quality of English Language

n/a

Round 2

Reviewer 2 Report

Comments and Suggestions for Authors

I made that point previously, but maybe I wasn’t clear enough or too gentle,so I have to emphasize that the Picrust after 16S metabarcoding can help predict functional patterns based on a generic phylogenetic tree (you need to clarify the approach in your Method). However, without clear experimental approaches, these predictions remain purely speculative and should not be overinterpreted. I am sorry but one reviewer made the same mistake and has encouraged the authors to develop even more superficial discussions that lack experimental support.

Something I missed initially – I thought it was a typo – was the use of "constrained principal coordinate analysis (CPCoA). This ordination method is really unconventional in the field, and after researching it, I assumed the authors likely meant "constrained analysis of principal coordinates (CAP)." First, they need to clarify their abbreviation and specify the exact method they used. Secondly, and this is more important, they must provide a clear rationale for using this method over traditional ones. I really don't get the mathematical justification for rda analyses here, and to me it seems superfluous in the context of gut microbiome studies. Authors added "CPCoA adds grouping information to the analysis of PCoA to find the plane that can explain the differences between groups to the greatest extent under self-defined grouping conditions ". What does it mean exactly? That said vegan.cca  relates to a CCA ordination method, which doesn't seem corresponding to visual graphic here. From everything, the separation among the three groups appears minimal. The choice of ordination method makes it seem like there is a significant shift in the 3 microbiome community, which I doubt is the case. I don't think taxonomy alone is the key, and I would not be surprised if there is minimal shift in 16S among the three groups, so I would prefer not to be fooled by a visual representation suggesting otherwise.

There is nothing like " ANOVA correlation " . If the presented p-value correspond to ANOVA test, then it is unecessary to add the "a" on your boxplots.

Figure 3. Add edgeR in your Figure legend explanation.

Round 3

Reviewer 2 Report

Comments and Suggestions for Authors

Dear authors, I find this revised version addresses my specific queries satisfactorily. The PCoA ordination clarifies that, despite no significant taxonomic changes, there may be alterations in minor taxa and potential shifts in microbial functions. Congratulations for an interesting study in the field.